# Bringing Preventive RSV Monoclonal Antibodies to Infants in Low- and Middle-Income Countries: Challenges and Opportunities

**DOI:** 10.3390/vaccines9090961

**Published:** 2021-08-28

**Authors:** Jintanat Ananworanich, Penny M. Heaton

**Affiliations:** 1Bill & Melinda Gates Medical Research Institute, Cambridge, MA 02139, USA; jananworanich@outlook.com; 2Amsterdam Medical Center, Department of Global Health, University of Amsterdam, 1105 BP Amsterdam, The Netherlands

**Keywords:** RSV, lower respiratory tract infection, monoclonal antibody, LMIC

## Abstract

Respiratory syncytial virus (RSV) is the leading cause of acute lower respiratory tract infections (LRTIs) in infants. Most deaths occur in infants under 3 months old, and those living in low and middle-income countries (LMICs). There are no maternal or infant RSV vaccines currently approved. An RSV monoclonal antibody (mAb) could fill the gap until vaccines are available. It could also be used when a vaccine is not given, or when there is insufficient time to vaccinate and generate an antibody response. The only currently approved RSV mAb, palivizumab, is too costly and needs monthly administration, which is not possible in LMICs. It is imperative that a safe, effective, and affordable mAb to prevent severe RSV LRTI be developed for infants in LMICs. Next generation, half-life extended mAbs in clinical development, such as nirsevimab, show promise in protecting infants against RSV LRTI. Given that a single dose could cover an entire 5-month season, there is an opportunity to make RSV mAbs affordable for LMICs by investing in improvements in manufacturing efficiency. The challenges of using RSV mAbs in LMICs are the complexities of integrating them into existing healthcare delivery programs and surveillance systems, both of which are needed to define seasonal patterns, and monitor for escape mutants. Collaboration with key stakeholders such as the World Health Organization and Gavi, the Vaccine Alliance, will be essential for achieving this goal.

## 1. Introduction

Respiratory syncytial virus (RSV) is a global disease that places a high and under-appreciated burden on the economy and healthcare services. It is the leading preventable cause of acute lower respiratory tract infection (LRTI) deaths in children 6 months of age or younger; most deaths occur in low- and middle-income countries (LMICs) [1]. 

The World Health Organization (WHO) Product Development for Vaccines Advisory Committee (PDVAC) indicates that RSV immunoprophylaxis with vaccines and monoclonal antibodies (mAbs) are priority interventions. There is currently no approved vaccine against RSV. RSV mAbs are proven to prevent RSV LRTI in preterm infants and are currently being used in infants at high risk for severe RSV disease. However, the only type approved is not accessible and affordable for widespread use in LMICs. The high rate of RSV infection, especially among infants and young children, underscores the need for safe, effective, and affordable prevention of RSV disease.

## 2. RSV’s Burden of Disease

Most children acquire an RSV infection by the time they are 2 years old, causing a mild upper respiratory tract infection within 4 to 6 days after infection. Population-based serologic studies suggest that seroconversion from primary RSV infection occurs in about 30% of infants by 6 months of age, 50% by age one, and 100% by age two. [2,3]. However, in some children it leads to a more severe illness, such as bronchiolitis or pneumonia. All infants are at risk of RSV-associated acute LRTI, but the highest risk is in the first 6 months of life, and in particular, the first 3 months. During this early infancy period, infants have narrow airways that are easily obstructed by inflammatory infiltrates, and may have low titers of maternally transferred neutralizing antibodies against RSV [4]. Infants who are born prematurely or have chronic lung disease (CLD) or congenital heart disease (CHD) are at the greatest risk of acute LRTI. Gestational age and birth weight are independent risks for hospitalization [5].

In 2015, RSV was estimated to cause 33.1 million new episodes of acute LRTI in children under 5 years old. Among 3.2 million hospital admissions in this age group, 1.4 million hospitalizations were in children younger than 6 months. Healthy term infants 3 months of age and younger account for more RSV-associated hospitalizations than any other age group [6]. The hospital admission rate from RSV in LMICs is 15.9 per 1000 neonates per year. Incidence rates are three times greater in preterm than term infants [1]. Globally, there are approximately 120,000 RSV-associated acute LRTI deaths among children <5 years of age each year [7]. RSV LRTIs account for a third of all LRTIs [1]. Most deaths occur in LMICs. Approximately 45% of in-hospital deaths due to RSV-associated acute LRTIs occur in children younger than 6 months (median ages of RSV-related deaths are between 4 and 7 months [1,8].

Preliminary data from 2019 suggest that in several low-income countries, 6–10% of all deaths in infants up to 6 months of age may be associated with RSV [9]. Data from the Child Health and Mortality Prevention Surveillance (CHAMPS) and ongoing community mortality surveillance platforms in Zambia and India indicate that RSV is associated with approximately 7% of all deaths in infants 7 days to 6 months of age. This suggests a potentially significant number of preventable RSV deaths in young infants. Data from Zambia indicate close to 70% of this RSV-associated mortality in young infants occurs in the first 3 months of life [10]. There is substantial underreporting of community, non-hospital deaths and non-respiratory RSV deaths (e.g., sepsis), indicating that infants in this age group in LMICs constitute a critically important target group for prevention.

## 3. RSV’s Molecular Characteristics

RSV is a negative-stranded RNA virus in the *Pneumoviridae* family. It has 11 known proteins, including envelope spikes (SH, G and F), inner envelope proteins (M), non-structural proteins (NS1, NS2) and ribonucleocapsid complexes (N, P, L, M2-1, M2-2). RSV has two subtypes, A and B, and these are typically determined by G sequences. F and G proteins are the main targets for vaccine and mAb development because of their direct involvement with infectivity and RSV disease.

The RSV F protein (involved in viral entry), is a superior target to the G protein (involved in attachment to host cell) because of its conserved and highly neutralization-sensitive epitopes. G is an attachment glycoprotein with membrane and secreted forms. The RSV F protein is essential for virus entry into the host cell and mediates membrane fusion of the virus particle and the target cell. Fusion of infected cells with adjacent cells results in the formation of the characteristic syncytia. During membrane fusion, the metastable prefusion (pre-F) undergoes conformational changes to a stable post-fusion F (post-F). There are six antigenic sites (Ø, I, II, III, IV, and V) between pre-F and post-F (Figure 1) [11]. Sites Ø and V are pre-F sites, whereas site I is a post-F site. Site II and site IV are shared by both the pre-F and post-F. Neutralizing antibodies for pre-F are more potent than those targeting post-F due to pre-F being a highly neutralization-sensitive region [12]. The discovery of the pre-F conformation and stabilization to present a fraction of neutralizing-sensitive surfaces as an immunogen has transformed RSV vaccine development and fueled next generation vaccines capable of inducing highly neutralizing polyclonal antibody responses. The identification of the antigenic sites on pre-F has advanced therapies such as vaccine and mAb development to reduce the burden of RSV disease [13].

## 4. Natural Immunity to RSV

Development of a therapy to prevent or reduce RSV infection is ongoing, building in part, on research related to natural RSV immunity. Previous research has shown that RSV-naïve infants who receive higher levels of neutralizing antibody from their mothers transplacentally and potentially via breastmilk are less susceptible to RSV infection [14,15]. Moreover, infants with ability to generate strong RSV-specific humoral and cellular immune responses during primary RSV infections tend to have faster recovery and less severe RSV [16]. Nevertheless, RSV-naïve infants typically generate innate and adaptive immune responses against RSV that are suboptimal and short-lived. This could be due to their developing and immature immune system, immune tolerance, and RSV-driven escape from host immunity (e.g., downregulation of type I interferon response, immune modulation of secretory G protein, and spontaneously flipping into post-F confirmation to avoid neutralization) [11]. Such original antigenic failure may imprint a long-lasting suboptimal adaptive immune response to RSV. Therefore, natural immunity to RSV is insufficient, and re-infection occurs throughout adulthood, although with milder disease severity, until immune senescence and dysregulated responses occur in the elderly, resulting in an increased risk of severe RSV disease [17].

Direct immunization in RSV-naïve infants offers a unique opportunity to elicit long-lasting, effective innate, humoral, and cellular immune responses at initial exposure to the virus. The goal of maternal immunization is to boost polyclonal anti-RSV neutralizing antibody titers for antenatal transfer to the infant, whereas RSV mAb blocks viral entry by targeting a specific antigenic site on the F protein.

## 5. RSV Prevention Strategies

All infants, both full-term and preterm, should be protected from RSV. The most important thing for LMICs is to prevent severe RSV LRTI in infants who are ≤6 months of age, and entering their first RSV season. Ideally, the protection should cover a typical 5-month RSV season.

As shown in Figure 2, there are three main strategies being studied to protect infants during this vulnerable period. The priorities are developing maternal RSV vaccines and infant RSV mAbs to provide immediate protection to young infants, whereas active infant immunization is focused on protecting infants after 6 months of age as maternal or passive antibody titers wane [18]. Achieving protective efficacy via active immunization may prove difficult for infants less than 3 months of age.

There is a rapidly expanding landscape of RSV vaccine development. There are more than 20 RSV vaccines in clinical development, including live attenuated/chimeric, protein-based, nucleic acid, and recombinant vectored vaccines [19]. Two RSV prefusion F protein subunit maternal vaccines are in phase 3 with results anticipated in 2023/2024, by GlaxoSmithKline [20] and Pfizer [21].

Maternal immunization aims to boost the generation of high potency neutralizing antibodies that are transferred via the placenta to the fetus during the second or third trimester. This would protect infants during the first 3 to 6 months of life. The WHO Preferred Product Characteristics (PPC) for RSV vaccines include a one dose regimen using a well characterized platform with a favorable safety profile in pregnancy, and greater than 70% vaccine efficacy against confirmed severe RSV disease in the offspring from birth to age 4 months or more [22]. For maternal RSV vaccine development, the WHO has defined severe RSV LRTI as polymerase chain reaction (PCR)-confirmed RSV infection with all of the following: respiratory infection (cough or difficulty breathing), LRTI defined as fast breathing by WHO criteria or SpO2 < 95%, and ≥ one of the following features of severe disease: SpO2< 93% and/or lower chest wall in-drawing [23].

The only completed phase 3 vaccine trial is the Novavax maternal RSV F nanoparticle vaccine trial [24]. This trial enrolled 4636 women in their third trimester of pregnancy. The primary endpoint, reduction in medically-attended RSV LRTI at 90 days, was not met based on the pre-specified success criterion, which had a lower bound of the 97.52% confidence interval (CI) of ≥30% (vaccine efficacy (VE) 39% (97.52% CI of −1 to 63.7)). However, the estimate of VE for reduction of LRTI with severe hypoxemia from RSV was higher, at 48.3% (95% CI of −8.2, 75.3%). Moreover, the vaccine did show benefits in reducing hospitalizations from RSV-associated LRTI (VE 44.4%, 95% CI, 19.6 to 61.5), and from all-cause LRTI (VE 27.8%, 95% CI 4.8 to 45.3).

Administration of a mAbs directly to the infant is a proven RSV preventive therapy [25] that not only bridges the gap before vaccines are available, but will also compliment vaccines after they become available for infants who require immediate protection. Refer to Table 1 for a list of completed and ongoing phase 2 and 3 trials of RSV mAb, and to Section 6 for details regarding the approved and candidate mAbs to date. The mAbs would only need to be used in infants 6 months of age and younger, or those at risk of LRTIs during the local RSV season, which would reduce the overall national program cost of RSV disease prevention. Palivizumab is currently the only licensed, short-acting mAb, and is approved for preterm infants, and infants with CLD and CHD. There are several half-life extended RSV mAbs in development, including at least four mAbs in early development and two in late-stage development [19]. Critical success factors intrinsic to a candidate mAb include high potency; a high barrier to mAb-resistant escape mutants; product safety, stability, and adequate pharmacokinetics; and adequate duration of protection. The WHO PPC for RSV mAbs include prevention of severe RSV disease in infants with a one-dose intramuscular or subcutaneous regimen that can be given as a birth dose or at any healthcare visit during the first 6 months of life. The mAb should have safety and reactogenicity profile comparable to other WHO recommended vaccines and have at least 70% efficacy against RSV-confirmed severe disease (due to both RSV A and B subtypes) for 5 months following administration. The mAb should be accessible and affordable to LMICs, and licensed by national regulatory authorities [22].

The strategy of immunizing infants is more suitable for protecting infants >3 months old, and children. There is insufficient time to vaccinate to generate an effective and protective immune response in infants younger than 3 months with a developing immune system. It would likely require multiple vaccine doses and several months to generate protective immunity, which could be long-lasting. Moreover, studies with formalin-inactivated vaccines in the 1960s suggested an enhancement of RSV disease post-vaccination in the young infants who were RSV seronegative before vaccination [26]. This was likely due to aberrant responses with induction of delayed type cellular responses [27], and non-neutralizing antibody responses leading to immune complex deposition and complement fixation in the small airways [28]. It should be noted that passively administered serum from formalin-inactivated RSV vaccinated rodents was not associated with enhanced pathology in the pups that received the sera and subsequently challenged with RSV [29]. Much has been learned since then, and current studies are focused on different types of vaccines that induce high titers of neutralizing antibodies, such as live attenuated, recombinant vectored, and nucleic acid vaccines, and they include seropositive infants and children [19].

## 6. RSV Monoclonal Antibody Studies

There is one approved RSV mAb, palivizumab, a prophylactic against LRTI caused by RSV that has been used for the past two decades. Palivizumab is an RSV F-specific immunoglobulin G monoclonal antibody (Synagis^®^, Swedish Orphan Biovitrum AB, Stockholm, Sweden) that has a half-life of around 20 days and requires monthly injections. This mAb, mainly used in high-income countries, is cost prohibitive for LMICs, and has limited approval for infants. It is approved by the Food and Drug Administration (FDA) for high-risk children, including (1) preterm infants born at ≤ 35 weeks gestational age and who are ≤ 6 months old at the beginning of RSV season; (2) children with bronchopulmonary dysplasia (BPD) that required medical treatment within the previous 6 months and who are ≤ 24 months old at the beginning of RSV season; and (3) children with hemodynamically significant CHD and who are ≤24 months old at the beginning of RSV season. Among preterm infants, the American Academy of Pediatrics (AAP) issued updated guidance in 2014 to limit its recommendation for palivizumab to preterm infants born before 29 weeks gestational age who are ≤12 months old, who would most likely benefit from this prophylaxis [42].

After the 2014 AAP policy, the overall palivizumab use declined and RSV-associated hospitalization in the US increased in infants < 6 months old, who had a gestational age of 29 to 34 weeks [43]. Similar increases were also observed in the Province of Quebec, Canada when palivizumab was withdrawn for use in late preterm infants [44]. These data support the need for a RSV immunoprophylaxis strategy for all young infants. Given the altered RSV epidemiology during COVID-19 pandemic—delayed seasonal onset and increases in inter-seasonal RSV activity—the AAP has issued a strong recommendation for palivizumab to be used in eligible patients living in regions that are experiencing high rates of RSV circulation [45].

Motavizumab, a derivative of palivizumab with higher affinity for RSV, was not pursued by MedImmune for licensure after the phase 3 trials in high-risk children failed to show clear superiority to palivizumab, and noted a trend for a higher rate of skin rashes [30,46]. Suptavumab (developed by Regeneron Pharmaceuticals, Inc., Tarrytown, NY, USA), given as a 2-dose regimen, 8 weeks apart, did not meet trial efficacy endpoints in phase 3 due to its lack of efficacy against the predominantly circulating RSV B strains, and development was discontinued [32].

Two RSV mAbs are in late-stage clinical development, MEDI8897 (nirsevimab, Astra Zeneca, Gaithersburg, MD, USA) and MK-1654 (Merck, Kenilworth, NJ, USA). Nirsevimab binds to the antigenic site Ø of RSV prefusion F, which is highly sensitive to neutralization [47]. MK-1654 binds to antigenic site IV of RSV pre- and post-fusion F protein, which is considered highly conserved [48]. These mAbs have broad neutralization activity against laboratory- and clinical-circulating strains of RSV A and B. Both have an engineered Fc domain with a half-life extension crystallizable fragment domain M252Y/S254T/T256E (YTE) mutation (European Union numbering) that prolongs the half-life by 3-fold compared to palivizumab, to about 70 days, due to enhanced binding to FcRn that impedes antibody clearance [49].

Nirsevimab received FDA breakthrough status and the European Medicines Agency (EMA) PRIME (priority medicines) status for its phase 2b trial, which enrolled premature infants 29 to 34 weeks in gestational age who were ≤1 year old. After a single dose of nirsevimab 50 mg via intramuscular (IM) injection, the incidence of RSV-associated medically-attended LRTI (MALRTI) was reduced by 70%, and that of hospitalizations by 78%, compared to placebo [34].

The ongoing phase 3 study in term and late preterm infants ≥35 weeks gestational age [37] has met its primary endpoint of reducing RSV LRTIs [38]. There is also an ongoing phase 2b/3 in preterm infants ≤ 35 weeks gestational age with either CLD or CHD or BPD [36]. These late-stage studies employ weight-based dosing (50 mg IM for infants weighing < 5 kg (kilogram) and 100 mg IM for those weighing ≥ 5 kg [50].

The MK-1654 program has begun recruitment into their phase 2b/3 trail in healthy preterm and full-term infants, ≤1 year of age, to evaluate a single IM administration of MK-1654 for preventing RSV-associated MALRTI [41]. The phase 2a study of safety and PK has completed recruitment, and monitoring is ongoing [39]. A phase 2a, controlled human challenge model of MK-1654 in healthy participants was completed, and results as of May 2021 show antiviral activity against RSV [40].

The safety data, to date, from nirsevimab and MK-1654 studies in adults and infants, have been favorable. The overall safety profile in phase 1 studies in adults, with dosing 5 to 10 times the target infant doses, is similar to that of placebo [51,52]. In the phase 2b study, in which approximately 900 infants received nirsevimab, no anaphylaxis or other notable hypersensitivity reactions were observed. There were no serious adverse events related to the mAb [34]. For both mAbs, anti-drug antibodies (ADA) post-mAb administration were infrequent and were not associated with adverse events. In the phase 1b/2a nirsevimab study, ADA was associated with lower drug exposure, but only between days 150 to 360 in some infants [6].

The current candidate mAbs in development are targeted primarily for high-income countries, with costs likely to be too high to allow affordable global access to the product, unless successfully addressed by other mechanisms, such as tiered pricing. The Bill and Melinda Gates Foundation (BMGF) has supported the preclinical development of a candidate mAb, RSM01, which is being advanced to clinical development by the Bill and Melinda Gates Medical Research Institute (Gates MRI), a non-profit biotechnology institution. RSM01 is a fully human IgG1 mAb that targets site Ø of the pre-F of RSV and has YTE half-life extended mutations. The BMGF has the primary aim of developing this product for LMICs.

## 7. Challenges and Opportunities in Bringing RSV Monoclonal Antibody to LMICs

### 7.1. RSV Seasonality and Timing of RSV mAb Dosing

A successful and cost effective implementation will rely on the ability to administer the mAbs at the beginning of the infant’s first RSV season. This assumes that a half-life extended mAb could protect infants against severe RSV disease through a typical 5-month RSV season, and that in the LMIC setting, only one dose per infant will be given during the first 6 months of life. In countries with clear RSV seasonality, the mAb could be given at birth or during a scheduled immunization visit, whichever is closest to the beginning of RSV season. A birth dose is ideal as it negates missed opportunities, particularly in areas with poor return rates for immunization visits. However, many LMICs have tropical climates where RSV seasonality may be less pronounced compared to temperate climate countries. A year-round RSV pattern, predominantly seen in countries near the equator, poses a challenge to implementing mAbs.

A recent mathematical modeling study to evaluate seasonality versus a year-round approach for mAb implementation in infants ≤ 4 months of age utilized surveillance data from 52 LMICs across Asia, Africa, Eastern Europe, South America, and the Middle East. The majority of these LMICs (75%) had clear RSV seasonality: more than 75% of annual RSV cases occurred in ≤5 months. In the countries with clear RSV seasonality, a seasonal approach that gives mAbs to eligible infants 1 to 3 months before onset of the RSV season achieved high efficacy without substantially compromising effectiveness compared to the year-round approach, while only utilizing about half of the mAbs (i.e., infants born outside RSV season would not require mAbs) [53]. In countries without clear RSV seasonality, or those without any RSV seasonality data, the mAbs could be given at birth, year-round. This approach could be simpler to implement and may also be considered for countries with RSV seasonality. A birth dose would reduce missed opportunities. However, this approach is likely less cost-efficient if infants are dosed outside the RSV season and the mAb duration of protection is short.

### 7.2. Escape Mutants

RSV, like other RNA viruses, possesses an error-prone RNA polymerase. Although the F protein, which is the target for most RSV mAbs in development, exhibits relative genetic and antigenic stability, there are potential concerns about the emergence of antibody resistant escape mutants. Resistance to RSV mAb could be due to the pre-existing resistant mutations in circulating RSV strains, as observed in the phase 3 suptavumab trial, or from selective pressure on the virus following mAb administration. Amino acid substitutions in the nirsevimab binding site have been observed in in-vitro experiments, and most did not impact viral replication [54]. However, nirsevimab-resistant mutants were observed in 2 of 25 breakthrough infections in the phase 2b study [34]. Molecular surveillance of the 2017–2018 F sequences from eight countries showed high-frequency polymorphisms at antigenic sites Ø and V of RSV B that included 100% detection of the nonconservative polymorphisms in suptavumab target site V (L172Q/S173L). Conservative polymorphisms in nirsevimab target site Ø (I206M/Q209R) [54] were detected in 77% of RSV B strains, but they retained susceptibility to neutralization by nirsevimab [55].

It is important for drug developers to conduct database analysis to determine if any new circulating strains exhibit potential for being resistant to their products. There is a risk that mAbs could drive resistant RSV strains (i.e., escape variants) by single point mutations, resulting in loss of binding and protection. However, after two decades of use, no known palivizumab target site II polymorphisms have been observed among 2017–2018 RSV strains. This is encouraging, albeit the limited use of palivizumab may not have put enough pressure on the virus to select for resistant strains.

### 7.3. Cost of Goods (COGs) and Scalability for LMICs

There is realistic potential for an RSV mAb to be scalable in LMICs. The current drug substance manufacturing processes for mAbs could reasonably yield COGs of under $100 per gram [56]. Assuming that a 50 mg dose would be sufficient to protect most infants <6 months of age, COGs could be less than $5 per dose. Multi-dose vial presentations will also reduce the relative costs of such drug products. With substantial investments toward improving manufacturing process and product presentation, cost, storage and distribution (including last-mile stability at ambient temperature) on par with some vaccines could be achieved.

### 7.4. Partnership with National and International Key Stakeholders

In order for countries to implement RSV mAbs as a preventive measure for infants, national level data of RSV incidence and the burden on healthcare utilization will be required. However, LMICs often lack such data due to limited infrastructure for conducting surveillance. A recent study showed that RSV burden data in children under 5 years old were only available from 8% of the population in low-income countries, compared to 63% in lower-middle-income countries, 80% in upper-middle-income countries, and 71% in high-income countries [57]. The WHO has built upon the influenza surveillance system to initiate global RSV surveillance in 25 countries to support the introduction of RSV immunization, should it become available. For countries with clear RSV seasonality, a few years of surveillance should be sufficient to inform a country’s seasonal RSV mAb administration strategy [53]. Underrepresented estimates of RSV-associated LRTI, hospitalization, and deaths will hinder the adoption of nationwide RSV preventive measures that could save millions of lives.

To facilitate delivery of an effective RSV mAb to LMICs, it would be important for RSV mAb to be on the WHO Pre-Qualification (PQ) list and the Essential Medication List (EML). Cancer therapeutic mAbs have been added to the EML in 2019, so there is precedent for this type of modality to be prequalified. The drug developers’ early engagement with different WHO committees (PDVAC, Technical Advisory Group (TAG) and Strategic Advisory Group of Experts (SAGE)) and Gavi, the Vaccine Alliance, will be necessary to obtain guidance on a clinical development plan and PQ process. It will be critical to seek guidance from WHO and other key stakeholders on the feasibility of conducting placebo-controlled studies in LMICs after nirsevimab or other RSV mAbs are approved, but are inaccessible and not yet standard of care [58].

A non-inferiority trial of a new RSV mAb against nirsevimab would require a very large sample size that may not be feasible. For example, assuming a 1% severe RSV disease rate in the nirsevimab group, a sample size of 140,101 infants/group (80% power) or 187,654 infants/group (90% power) would be needed to show a lower bound on the relative risk >0.9 (i.e., control incidence is lower than RSM01 by at most 10% with 1-sided 97.5% CI). As there are more available data from RSV mAb or vaccine trials, a neutralizing antibody titer may be established as a correlate of protection against RSV, which could allow for a smaller phase 3 to be conducted using biomarker endpoints.

### 7.5. Complimentary Roles of RSV mAb and Maternal RSV Vaccines in Preventing Severe RSV LRTI in Infants

Maternal RSV vaccines require administration during a specific gestational window of 24 to 36 weeks in women who seek antenatal care. Premature birth and late vaccination can result in lower antibody titers and shorter durations of protection in infants. Moreover, infants who have additional risk factors for severe LRTI (e.g., preterm, CLD, or CHD) may require protection beyond the first 3 to 6 months of life, when placentally transferred Ab has waned. In these instances, RSV mAbs could be administered during a wider window to provide immediate protection for infants. A modeling study projected both maternal RSV vaccine and mAb to be independently impactful and cost-effective in LMICs when administered to infants < 6 months old. Compared to no intervention, a maternal vaccine or mAbs (birth dose) are projected to avert 25% or 55% of RSV-related deaths, respectively, based on similar assumptions for efficacy (minimum 30% for 4 months and maximum 90% for 6 months) [59].

### 7.6. Lessons Learned from COVID-19

The COVID-19 pandemic has propelled innovations in clinical trials that could be leveraged for RSV mAb studies. In our own experience conducting a COVID-19 treatment trial [60], an accelerated timeline was achieved for all steps, from study concept to regulatory review, and study start. We demonstrated successful clinical operation using a 100% remote trial model that recruited participants from across the US via a meta-site. This type of process and other innovations (e.g., wearables and a self-collecting blood sample device) could facilitate collection of data and inclusion of participants regardless of where they live, thereby alleviating the travel burden on families and children in RSV mAb trials for post-mAb administration follow-up visits.

## 8. Conclusions

RSV is the most common infectious cause of acute LRTI in young children. Death rates are higher in infants < 3 months old and those living in LMICs. An efficacious RSV immunization product provided to all infants in the first 6 months of life as they enter their first RSV season could have significant public health impact by preventing LRTI-associated hospitalizations and deaths. It might also reduce all-cause LRTIs, as shown in the Novavax maternal RSV vaccine trial. There are dozens of maternal and infant RSV vaccines being studied, but none have reached approval. Administering mAbs directly to the infant is the only proven RSV preventive therapy to date. Several next-generation, half-life extended mAbs in development have shown promise as single injections to protect infants from RSV-associated LRTIs for the entire 5-month RSV season. Such RSV mAbs could be scalable in LMICs because of their low dose requirements, and potential for multi-dose vial presentation and further improvements in manufacturing processes.

## Figures and Tables

**Figure 1 vaccines-09-00961-f001:**
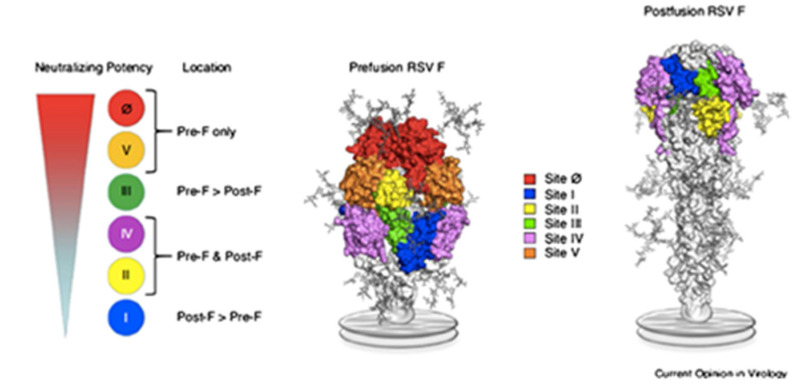
Locations of 6 antigenic sites on the prefusion (**left**) and postfusion (**right**) structures of RSV F glycoprotein and their neutralizing potencies. Reprinted by permission from Springer Nature. Clin Microbiol & Infect Dis. Human respiratory syncytial virus: pathogenesis, immune responses, and current vaccine approaches. Taleb, et al., 2018 [11]. copyright permission from Springer.

**Figure 2 vaccines-09-00961-f002:**
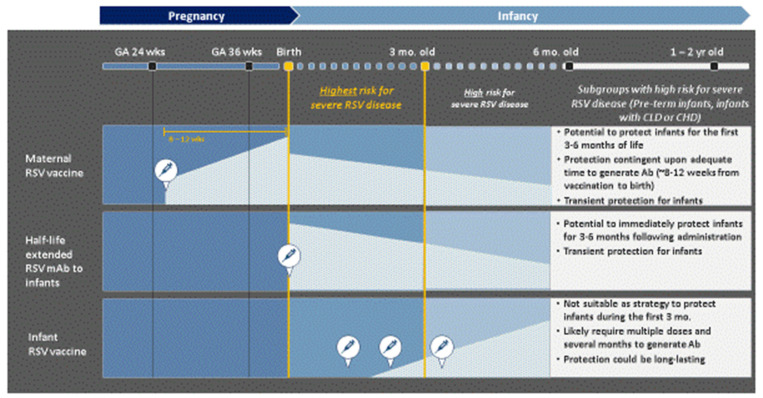
RSV-preventing strategies for infants.

**Table 1 vaccines-09-00961-t001:** Completed and ongoing phase 2 and 3 trials of RSV mAbs.

mAb	Stage	Populations and Outcomes
Palivizumab Commercial product	FDA- and EMA-approved in 1998 (*n* = 1502) [25]	Preterm <29 weeks gestational age (GA), congenital heart disease (CHD) ≤1 year old, chronic lung disease (CLD) ≤ 2 years old Reduction in incidence of RSV hospitalization: − Palivizumab 48/1002 (4.8%) vs. Placebo 53/500 (10.6%), 55% reduction (95% CI 38–72), *p* < 0.001
Motavizumab Manufacturer voluntarily withdrew product from FDA submission (discontinued in 2010)	Cardiac study (*n* = 1248; 2005 to 2008) [30]	CHD ≤ 2 years old Reduction in incidence of RSV hospitalization: − Motavizumab (1.9%) vs. Palivizumab (2.6%), 25% reduction (RR 0.75, CI 0.34–1.59) Rash incidence: Motavizumab (10.2%) vs. Pali (7.2%) − Skin serious adverse events (SAEs): Motavizumab (1.3%) vs. Palivizumab (0.3%)
	Phase 3 (*n* = 2127; 2004 to 2007) [31]	Healthy term native Americans ≥36 weeks GA ≤ 6 months old Incidence of RSV hospitalization: − Motavizumab 21/1417 (1%) vs. Placebo 80/710 (11%), 87% reduction (RR 9.8, CI 7.4–12.2), *p* < 0.001 Hypersensitivity: Motavizumab (14.7%) vs. Placebo (12.3%), *p* = 0.14. − 8 SAEs in Motavizumab group: 6 hypersensitivity, 1 erythema multiforme, 1 skin erythema. 6/8 events were after 4–5 doses
Suptavumab (REGN2222)	Phase 3 (*n* = 1149; 2015 to 2017) [32,33]	Failed to meet Phase 3 endpoint. Program was discontinued in 2017 Preterm ≤35 weeks GA ≤ 6 months Rates of medically-attended RSV LRTI (MALRTI): − One dose Suptavumab (7.7%) vs. 2 doses Suptavumab (9.3%) vs. Placebo (8.1%) Different by RSV subtypes for reductions of RSV in-patient and outpatient visits for Suptavumab vs. Placebo: − RSV A: One dose: 62.1% (95% CI −4.9 to 86.3); 2 doses: 61.4% (95% CI −6.6 to 86) − RSV B: One dose: −36.3% (95% CI −155.6 to 27.2); 2 doses: −68.7% (95% CI −208 to 7.5)
Nirsevimab (MEDI8897)	Phase 2b (*n* = 1453; 2016 to 2018) [34,35]	FDA granted breakthrough status; EMA granted Prime status Preterm 29–34 weeks GA; ≤ 1 year old entering their first full RSV season Incidence of MALRTI: − Nirsevimab 25/969 (2.6%) vs. Placebo 46/484 (9.5%). Reduction of 70.1% (95% CI 52.3–81.2%), *p* < 0.001. Incidence of RSV hospitalization − Nirsevimab 8/969 (0.8%) vs. Placebo 20/484 (4.1%). Reduction of 78.4% (95% CI 51.9–90.3), *p* = 0.001
	Phase 2b/3 (*n* = 925; 2019 to 2022) [36]	Preterm ≤ 35 weeks GA, CLD or CHD≤ 1 year old randomized to nirsevimab vs. placebo Recruitment is complete. Estimated study completion date in November 2022
	Phase 3 (*n* = 3000; 2019 to 2023) [37,38]	Term and preterm (≥35 weeks GA) ≤ 1 year old entering their first RSV season Study met primary endpoint of reduction of RSV-MALRTI. Estimated study completion date in March 2023
MK-1654	Phase 2a (*n* = 180; 2018 to 2022) [39]	Term and preterm (≥29 weeks GA) ≤ 8 months old randomized to 4 different doses of MK-1654 and placebo Recruitment is complete. Estimated study completion date in September 2022
	Phase 2a (*n* = 80; 2019 to 2020) [40]	Human challenge study with healthy adults randomized to 4 different doses of MK-1654 and placebo Non-statistically significant reductions were observed with MK-1654 compared to placebo for ‘Area under the Viral Load-time Curve’ and percentage of participants with symptomatic RSV infection post viral inoculation
	Phase 2b/3 (*n* = 3300; 2021 to 2025) [41]	Term and preterm ≥35 weeks gestational age ≤1 year old entering their first RSV season Recruitment began in April 2021 with estimated study completion date in January 2025

## Data Availability

Not applicable.

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
