# Peer review of "Bringing Preventive RSV Monoclonal Antibodies to Infants in Low- and Middle-Income Countries: Challenges and Opportunities"

_vaccines, 2021, doi:10.3390/vaccines9090961_

Round 1

Reviewer 1 Report

The review article, “Bringing preventive RSV monoclonal antibody to infants in  low- and middle-income countries: challenges and opportunities” by Jintanat Ananworanich and Penny Heaton is a thorough summary of the history and the current state of affairs for RSV prevention in infants. The article is well written and provides a balanced review of the opportunities and challenges in making an RSV mAb available to infants living in low- and middle-income countries. 

Major issues

None

Minor issues

Line 38: The high burden of RSV infection.  Recommend changing to the high rate of RSV infection as readers often get confused between RSV infection and RSV disease.

Line 41: Recommend changing subtitle from RSV burden to RSV burden of disease

Line 42-60: It would be helpful if the authors could add information on infection rates in the first 2 years of life in addition to hospitalization rates.  For example, hospitalization rates are highest in infants in under 3 months of age, but this is the same group that should have the highest level of maternal antibodies.  What is know about sero-positivity to RSV by age in the first 2 years of life?

Line 105: Recommend adding immature in front of developing immune system

Line 135: Note that GSK recently stopped development of their RSV infant vaccine. https://www.fiercebiotech.com/biotech/gsk-drops-phase-2-viral-vector-rsv-vaccine-pediatric-ages

Line 160: ‘Refer to Error!’  looks like a reference is missing

Line 170: the WHO PPC for an RSV mAb has now been published. 

Line 186: Table 1.

Note: Motavizumab was not submitted for approval.  MedImmune voluntarily withdrew the submission and it was not rejected by the FDA.

Motavizumab Ph3 healthy term study was in native Americans [26] O'Brien KL et al. Efficacy of motavizumab for the prevention of respiratory syncytial virus disease in healthy Native American infants: a phase 3 randomised double-blind placebo-controlled trial. Lancet Infect Dis. 2015 Dec;15(12):1398-408. Risk factors and RSV disease incidence is different in Native Americans than in the general infant population.

Also noting that there was an additional Motavizumab study in high-risk children that showed that Motavizumab was non-inferior to Palivizumab (Motavizumab recipients had a 26% relative reduction in RSV hospitalization compared with palivizumab recipients, achieving noninferiority (Carbonell-Estrany X, Motavizumab for prophylaxis of respiratory syncytial virus in high-risk children: a noninferiority trial. Pediatrics. 2010 Jan;125(1):e35-51.) that should be referenced.

Note that Nirsevimab has been granted CDE breakthrough status in China in addition to FDA breakthrough status and EMA Prime status.

Reference [31] Nirsevimab Ph2/3 Clinicaltrials.gov No. NCT03959488. Enrollment numbers were updated on July 19, 2021.

Reference [32, 33] Study met primary endpoint of reduction of RSV-MALRTI not MALRTI

Line 207: Noting again that MedImmune withdrew Motavizumab application and it did not fail to secure FDA approval.

Line 250: Authors are making assumptions about costs and access that could be addressed by other mechanisms such as tiered pricing.   

Line 299 [4847] formatting issue on references 47, 48

Line 305.  This is encouraging, albeit the limited use of palivizumab. I think the authors meant to finish the sentence with something to the effect of “…albeit the limited use of palivizumab may not have put enough pressure on the virus to select for resistant strains”

Line 374: type ‘under studied’ should be ’being studied’

Author Response

Reviewer 1 comments

Minor issues

  1. Line 38: The high burden of RSV infection.  Recommend changing to the high rate of RSV infection as readers often get confused between RSV infection and RSV disease.

Response: We have modified the text accordingly.

  1. Line 41: Recommend changing subtitle from RSV burden to RSV burden of disease

Response: We have modified the text accordingly.

  1. Line 42-60: It would be helpful if the authors could add information on infection rates in the first 2 years of life in addition to hospitalization rates.  For example, hospitalization rates are highest in infants in under 3 months of age, but this is the same group that should have the highest level of maternal antibodies.  What is known about sero-positivity to RSV by age in the first 2 years of life?

Response: Thank you for this comment. We have included additional text on the seroconversion rates during the first 2 years of life (line 43) as well as stating low titers of maternally transferred NAb as a possible reason for risk of acute LRTI in young infants (line 50).  Please note that under the RSV natural immunity section (line 107), we stated that RSV naïve infants who receive higher levels of NAb from mothers are less susceptible to RSV infection.

  1. Line 105: Recommend adding immature in front of developing immune system

Response: We have modified the text to “developing and immature immune system”.

  1. Line 135: Note that GSK recently stopped development of their RSV infant vaccine. https://www.fiercebiotech.com/biotech/gsk-drops-phase-2-viral-vector-rsv-vaccine-pediatric-ages

Response: Thank you for noting the recent discontinuation of the GSK infant RSV vaccine.  However, the text refers to the Phase 3 GSK maternal RSV vaccine that is ongoing.  We have kept the text unchanged.

  1. Line 160: ‘Refer to Error!’  looks like a reference is missing

Response: This has been corrected.

  1. Line 170: the WHO PPC for an RSV mAb has now been published.

Response: Thank you for this update. We have included information from the WHO PPC accordingly.

  1. Line 186: Refer to Table 1.

Response: We would like to clarify this request.  Table 1 refers to mAb studies whereas line 186 refers to vaccines. 

  1. Motavizumab was not submitted for approval.  MedImmune voluntarily withdrew the submission and it was not rejected by the FDA.

Response: Table 1 and body of review have been updated accordingly.

  1. Motavizumab Ph3 healthy term study was in native Americans [26] Risk factors and RSV disease incidence is different in Native Americans than in the general infant population.

Response: We have modified the text accordingly.

  1. Also noting that there was an additional Motavizumab study in high-risk children that showed that Motavizumab was non-inferior to Palivizumab (Motavizumab recipients had a 26% relative reduction in RSV hospitalization compared with palivizumab recipients, achieving noninferiority (Carbonell-Estrany X, Motavizumab for prophylaxis of respiratory syncytial virus in high-risk children: a noninferiority trial. Pediatrics. 2010 Jan;125(1):e35-51.) that should be referenced.

Response: We have included this reference and modified the text to state “Phase 3 trials in high-risk children”.

  1. Note that Nirsevimab has been granted CDE breakthrough status in China in addition to FDA breakthrough status and EMA Prime status.

Response: Thank you for the suggestion. We would like to keep the text as is. For the purposes of this manuscript, we are limiting mention of regulatory activities to those involving stringent regulatory authorities as recognized by the World Health Organization. In addition, we think that the information on the special status provided by the FDA and EMA for nirsevimab sufficiently highlights the importance of the development of this mAb for RSV prevention.

  1. Reference [31] Nirsevimab Ph2/3 Clinicaltrials.gov No. NCT03959488. Enrollment numbers were updated on July 19, 2021.

Response: Enrollment number for this trial in Table 1 has been updated.

  1. Reference [32, 33] Study met primary endpoint of reduction of RSV-MALRTI not MALRTI

Response: Text in Table 1 has been updated accordingly.

  1. Line 207: Noting again that MedImmune withdrew Motavizumab application and it did not fail to secure FDA approval.

Response: We have modified the text accordingly.

  1. Line 250: Authors are making assumptions about costs and access that could be addressed by other mechanisms such as tiered pricing. 

Response: Thank you for point this out. This has been added to the text.

  1. Line 299 [4847] formatting issue on references 47, 48

Response: These have been corrected.

  1. Line 305.  This is encouraging, albeit the limited use of palivizumab. I think the authors meant to finish the sentence with something to the effect of “…albeit the limited use of palivizumab may not have put enough pressure on the virus to select for resistant strains”

Response: Thank you for pointing this out. We have updated the text accordingly.

  1. Line 374: type ‘under studied’ should be ’being studied’

Response: We have modified the text accordingly.

Reviewer 2 Report

This well written manuscript provides an overview of the RSV burden with a focus on low-middle income countries and strategies to prevent moderate to severe RSV disease in infants. Other than palivizumab there are no other approved vaccine or mAb for prevention of RSV disease. A description of the new extended half-life mAbs in late phase development is provided with their use complementing a maternal immunization strategy.

I only have minor comments.

Line 87, “… post F (I and II in particular)…”. Please revise this statement. Suggest stating that Site II similar to site IV are shared by both the pre-fusion and post-fusion conformations. Site I is a post conformation site.

Figure 2. Suggest extending the figure by adding another column that is specific to the bulleted comments. Currently the bulleted comments fall under the subgroups with high risk for severe RSV, which makes it confusing. Another potential benefit for maternal immunization is protection of the mother against severe RSV disease, although this has not been documented.

Line 146. The definition of severe RSV LRTI does not appear correct because it states “…SpO2 <95% and pulse oximetry <93%”. It has to be one or the other but not both. A pulse oximetry of <93% is the same as SpO2<93%. Please revise.

Line 149, reference 20. Please provide the NEJM reference for Novavax phase 3 maternal immunization trial and not the clinicaltrials.gov reference. (Madhi et al. N Engl J Med. 2020 Jul 30;383(5):426-439. doi: 10.1056/NEJMoa1908380.  PMID: 32726529)

Lines 157-158. Please revise this section. Administration of a mAb directly to the infant at birth or at any time during the primary immunization series is a proven RSV prevention therapy…” Please provide a reference for this statement or revise it. In the U.S. palivizumab administration is based on the seasonality of the outbreak (normally November-March) and not around birth or the primary immunization series.

Line 160-161. I believe this was not intended to be left in the manuscript. “Refer to error! Reference source not found.”

Line 161, Incomplete sentence starting with “for a list of…”

Line 179-180 please note that reference 22 refers to delayed type cellular immunity. At that time TH-2 response had not been described. Either revise the phrase or the reference. Also it should be noted that passively administered serum from FI-RSV vaccinated rodents (mouse or cotton rat) was never associated with enhanced pathology in the pups that received the sera and subsequently challenged with RSV.

Line 212. Lower efficacy suggest there was efficacy against RSV/B isolates. Should consider restating “lack of efficacy” due to the rapid emergence of resistant mutations at antigenic site V. This issue is further commented in the escape mutants section. 

Author Response

Reviewer 2 comments

I only have minor comments.

  1. Line 87, “… post F (I and II in particular)…”. Please revise this statement. Suggest stating that Site II similar to site IV are shared by both the pre-fusion and post-fusion conformations. Site I is a post conformation site.

Response: We have modified the text accordingly.

  1. Figure 2. Suggest extending the figure by adding another column that is specific to the bulleted comments. Currently the bulleted comments fall under the subgroups with high risk for severe RSV, which makes it confusing. Another potential benefit for maternal immunization is protection of the mother against severe RSV disease, although this has not been documented.

Response: The last column includes bullets that pertain to the subgroups of infants with high risk for severe RSV disease. Since this article is focused on the prevention of RSV in infants, we would prefer to keep the figure as is.

  1. Line 146. The definition of severe RSV LRTI does not appear correct because it states “…SpO2 <95% and pulse oximetry <93%”. It has to be one or the other but not both. A pulse oximetry of <93% is the same as SpO2<93%.

Response: We have corrected the text accordingly. 

  1. Line 149, reference 20. Please provide the NEJM reference for Novavax phase 3 maternal immunization trial and not the clinicaltrials.gov reference. (Madhi et al. N Engl J Med. 2020 Jul 30;383(5):426-439. doi: 10.1056/NEJMoa1908380.  PMID: 32726529)

Response: The reference has been updated.

  1. Lines 157-158. Please revise this section. Administration of a mAb directly to the infant at birth or at any time during the primary immunization series is a proven RSV prevention therapy…” Please provide a reference for this statement or revise it. In the U.S. palivizumab administration is based on the seasonality of the outbreak (normally November-March) and not around birth or the primary immunization series.

Response: Thank you. The reviewer is correct. We had deleted “at birth or at any time during the primary immunization series” and included a reference to palivizumab.

  1. Line 160-161. I believe this was not intended to be left in the manuscript. “Refer to error! Reference source not found.”

Response: This has been corrected.

  1. Line 161, Incomplete sentence starting with “for a list of…”

Response: This has been corrected to “Refer to Table 1 for a list of.

  1. Line 179-180 please note that reference 22 refers to delayed type cellular immunity. At that time TH-2 response had not been described. Either revise the phrase or the reference. Also it should be noted that passively administered serum from FI-RSV vaccinated rodents (mouse or cotton rat) was never associated with enhanced pathology in the pups that received the sera and subsequently challenged with RSV.

Response: Thank you for pointing this out.  We have replaced type 2 with delayed type, and have included the text that the reviewer has suggested along with the reference.

  1. Line 212. Lower efficacy suggest there was efficacy against RSV/B isolates. Should consider restating “lack of efficacy” due to the rapid emergence of resistant mutations at antigenic site V. This issue is further commented in the escape mutants section.

Response: We have updated the text accordingly.

Additional changes made by the authors

Additional changes made by the authors

  1. Lines 114, 115, 116: Per Springer Nature license to reuse Figure 1, the following text in bold, was added to the Figure legend.

Figure 1: Location of 6 antigenic sites on prefusion (left) and postfusion (right) structure of RSV F glycoprotein and their neutralizing potency [11]. Reprinted by permission from Springer Nature. Clin Microbiol & Infect Dis. Human respiratory syncytial virus: pathogenesis, immune responses, and current vaccine approaches. Taleb, et al. 2018.

  1. Line 212: Information was added on AAP strong recommendation for palivizumab in light of the altered RSV epidemiology during COVID-19 pandemic